## RESEARCH ARTICLE

# Transient effect of melatonin treatment after neonatal hypoxic-ischemic brain injury in rats

**Hester Rijkje Berger[1,2], Axel K. G. Nyman[3,4], Tora Sund Morken[5,6], Marius Widerøe[3]***

**1** Department of Clinical and Molecular Medicine, Norwegian University of Science and Technology, Trondheim, Norway, **2** Department of Pediatrics, St. Olav University Hospital, Trondheim, Norway, **3** Department of Circulation and Medical Imaging, Norwegian University of Science and Technology, Trondheim, Norway, **4** Department of Neurology, St. Olav University Hospital, Trondheim, Norway, **5** Department of Neuromedicine and Movement Science, Norwegian University of Science and Technology, Trondheim, Norway, **6** Department of Ophthalmology, St. Olav University Hospital, Trondheim, Norway

* marius.wideroe@ntnu.no

## Abstract

Melatonin has potential neuroprotective capabilities after neonatal hypoxia-ischemia (HI), but long-term effects have not been investigated. We hypothesized that melatonin treatment directly after HI could protect against early and delayed brain injury. Unilateral HI brain injury was induced in postnatal day 7 rats. An intraperitoneal injection of either melatonin or vehicle was given at 0, 6 and 25 hours after hypoxia. *In-vivo* MRI was performed 1, 7, 20 and 43 days after HI, followed by histological analysis. Forelimb asymmetry and memory were assessed at 12–15 and at 36–43 days after HI. More melatonin treated than vehicle treated animals (54.5% vs 15.8%) developed a mild injury characterized by diffusion tensor values, brain volumes, histological scores and behavioral parameters closer to sham. However, on average, melatonin treatment resulted only in a tendency towards milder injury on $T_2$-weighted MRI and apparent diffusion coefficient maps day 1 after HI, and not improved long-term outcome. These results indicate that the melatonin treatment regimen of 3 injections of 10 mg/kg within the first 25 hours only gave a transient and subtle neuroprotective effect, and may not have been sufficient to mitigate long-term brain injury development following HI.

## Introduction

Perinatal hypoxia-ischemia (HI) is a significant cause of mortality and neurologic disabilities in late preterm and term neonates [1]. A lack of oxygen and glucose supply to the brain causes an impaired oxidative metabolism and a depletion of energy levels. This first energy failure initiates a cascade of toxic events including overstimulation of glutamate receptors and the formation of reactive oxygen species. After resuscitation, there is a latent phase with restored energy levels. Subsequently, after 6–48 hours, there is a secondary energy failure followed by a delayed injury cascade which is characterized by oxidative stress [2], inflammation and delayed cell death [3]. The development of brain injury can evolve further over months or years with ongoing inflammation and epigenetic changes [4].

**Data Availability Statement:** All data files and are available from the Norwegian Centre for Research Data database (DOI: http://dx.doi.org/10.18712/NSD-NSD2713-V1)

**Funding:** The study was funded by PhD grants provided by the Faculty of Medicine and Health

Sciences at the Norwegian University of Science and Technology (HRB and AKGN). The funders had no role in study design, data collection and analysis, decision to publish, or preparation of the manuscript.

**Competing interests:** The authors have declared that no competing interests exist.

Therapeutic hypothermia is now standard of care for term and late preterm infants with moderate-to-severe hypoxic-ischemic encephalopathy, but its efficacy is limited, especially in the presence of inflammation [5]. Additional therapies are therefore urgently needed. Melatonin is an endogenous neurohormone with pleiotropic neuroprotective effects and a favorable safety profile. Endogenous melatonin levels are low in neonates [6]. Preclinical and clinical evidence indicates short term treatment effects of melatonin [7,8]. Since HI brain injury develops over an extended period, it is of great importance to evaluate how melatonin affects long-term gray and white matter development after neonatal HI. However, only a few studies (with positive results) have investigated the long-term effects of melatonin treatment with the focus on behavioral assessment and end-point histology [9,10].

In this study we have used longitudinal Magnetic Resonance Imaging (MRI) and functional tests to study the effect of melatonin on brain injury development during the first 6 weeks after HI in neonatal rats. We hypothesized that melatonin treatment initiated directly after HI protects against early and delayed brain injury.

## Materials and methods

### Animals and drugs

Sprague Dawley rats were bred in the Comparative Medicine Core Facility at the Norwegian University of Science and Technology in Trondheim. Dams and their pups were kept on a 12:12 hour light-dark cycle with food and water ad libitum. All animal experiments were performed according to European Union and Norwegian regulations and guidelines for animal experimentation and approved by the Norwegian Animal Research Authority (Permit number 4975).

Melatonin, dimethyl sulfoxide (DMSO) and paraformaldehyde were purchased from Sigma-Aldrich (St. Louis, MA, USA / Irvine, UK), isoflurane from Baxter Medication Delivery (Oslo, Norway), bupivacain (5mg/ml) from AstraZenica (Oslo, Norway), phosphate buffered saline (PBS) from Fisher Scientific (Oslo, Norway) and phenobarbital from Farma Holding (Oslo, Norway).

### Hypoxic-ischemic brain injury

The Vannucci model [11] was used to induce unilateral HI brain injury to the right cerebral hemisphere. In short, the right carotid artery was identified and thermo-cauterized in 7 day old, anaesthetized rat pups of both sexes. After a recovery period, the pups were exposed to hypoxia (8% O2) for 105 minutes. Sham-operated littermates underwent the same procedure except for right carotid artery cauterization and hypoxia.

### Experimental groups

The number of pups in each litter was reduced to 10 before the Vannucci procedure. The animals were randomly assigned to HI (n = 30) or sham (n = 18) and one of the treatment conditions before surgery. Because the solubility of melatonin in water is low, DMSO was used as a solvent. To control for neurotoxic or neuroprotective effects of DMSO, we included a DMSO treated group. Thus, the following treatments were given: (1) a 4 mg/ml solution of melatonin (MEL, 10 mg/kg) in PBS with 5% DMSO (HI+MEL n = 11, sham+MEL n = 6), (2) PBS with 5% DMSO (HI+DMSO n = 11, sham+DMSO n = 6) or (3) only PBS (HI+PBS n = 8, sham +PBS n = 6). All animals received intraperitoneal injections at 3 time-points: immediately, 6 hours and 25 hours after the hypoxia. After the last day of scanning the animals were

euthanized with an overdose of phenobarbital and the brains were prepared for histological tissue examination by cardiac perfusion with paraformaldehyde (4%).

## In-vivo MRI acquisition

MRI was performed on a 7T magnet (Bruker BioSpec 70/20, Bruker BioSpin, Ettlingen, Germany) using a 86 mm volume resonator for RF transmission and a phased array mouse head (day 1 and 7 after HI) or rat head (day 20 and 43 after HI) surface coil for reception. During scanning the animals were anaesthetized with isoflurane (3% induction, 1.5% maintenance) in a mixture of $O_2$ and air (30% $O_2$). They were placed in a prone position on a water-heated bed. Depending on the age, the heads of the rats were fixated using a nose mask and styrofoam. The respiration and $O_2$ saturation were monitored.

ADC maps were obtained 1 day after HI. An echo planar imaging (EPI) sequence with Stejskal-Tanner diffusion weighting was acquired using 6 b-values (100/200/400/600/800/1000 s/mm$^2$) in 3 orthogonal directions, 1 b0 image, TE 30 ms, TR 3000 ms, 6 averages, Field of View (FOV) 17.2 x 15 mm2, acquisition matrix (MTX) 110 x 96, spatial resolution: 156 x 156 μm2, 9 slices of 1 mm slice thickness.

$T_2$-weighted ($T_2$-w) images were obtained on day 1 and 7 after HI using a Rapid Acquisition with Relaxation Enhancement (RARE) sequence; RARE factor 8, effective TE 70 ms, TR 3500 ms, 4 averages, FOV 20 x 20 mm$^2$, MTX 160 x 160, spatial resolution 125 x 125 μm$^2$, 15 slices of 1 mm thickness and acquisition time 4 min and 40 s. On day 20 and 43 after HI, the scan parameters were kept unchanged, but the number of averages and acquisition time were doubled to compensate for reduced sensitivity of the larger coil.

Diffusion Tensor Images (DTI) were obtained on day 7, 20 and 43 after HI (Fig 1) using an EPI sequence with Stejskal-Tanner diffusion weighting with a b-value of 750 s/mm$^2$ in 81 directions and 10 b0 images, TE 27.50 ms, TR 3000 ms, FOV 19.25 x 13.65 mm$^2$, MTX 110 x 78 reconstructed to 220 x 156, 12 axial slices of 0.75 mm thickness, 8 averages and 36 min and 24 s acquisition time.

## MRI analysis

ADC maps were calculated by fitting a mono-exponential model to the signal intensity of the different diffusion weighted images using MATLAB (version R2012b; The Math Works, Natick, MA, USA). MIPAV (version 6.0.1; NIH Center for Information Technology, Bethesda,

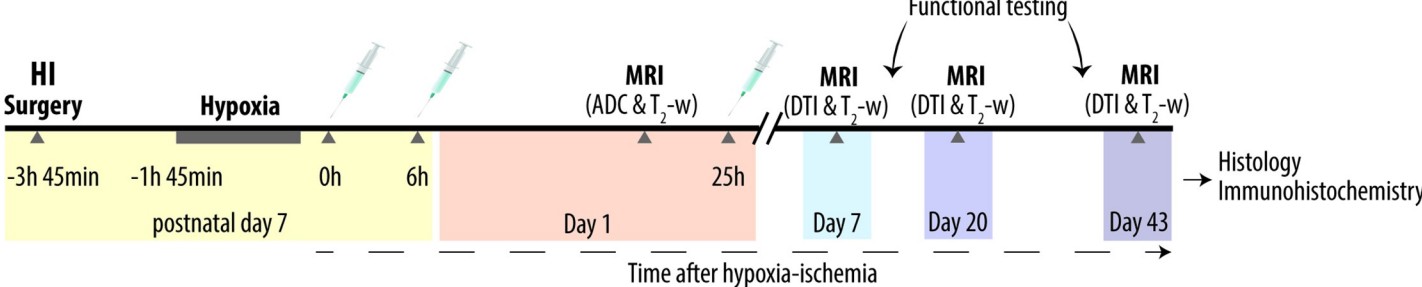

**Fig 1. Experimental design.** Postnatal day 7 rats were subjected to unilateral HI brain injury and received intraperitoneal injections of either melatonin 10 mg/kg dissolved in DMSO 5% and PBS or vehicle (DMSO 5% and/or PBS) at 0, 6 and 25 hours after hypoxia. Littermate controls were sham-operated. Within 1 day after HI, in-vivo MR imaging including ADC maps and $T_2$-w images was performed. $T_2$-w images and DTI were repeated at 7, 20 and 43 days after HI followed by histological and immunohistochemical analysis. Functional testing, including the cylinder rearing test and the novel object recognition test, was performed at 2 and 5–6 weeks after HI. ADC, apparent diffusion coefficient; DMSO, dimethyl sulfoxide; DTI, diffusion tensor imaging; HI, hypoxia-ischemia; MRI, magnetic resonance imaging; PBS, phosphate-buffered saline.

MD, USA) was used for further image analysis. Regions of interest (ROIs) were manually drawn in the caudate putamen, lateral and medial cortex, hippocampus and deep cerebral white matter, thalamus and basal ganglia in the right hemisphere and the mean ADC in each ROI was calculated.

$T_2$-w images were used for volumetric measurements. Volumes of interest were manually drawn in the images in the region between Bregma positions 3.24 and -7.32, including the telencephalon and diencephalon but excluding the olfactory bulbs and the mesencephalon. The four volumes represented cerebral tissue in the right hemisphere (ipsilateral to the lesion) with (i) an increased $T_2$ intensity, (ii) reduced $T_2$ intensity, (iii) intact tissue, and (iv) intact tissue in left hemisphere (contralateral to the lesion). Increased or reduced $T_2$ intensity in the ipsilateral hemisphere was defined as more than 2 SD above or below the mean $T_2$ intensity in the contralateral hemisphere. To derive an outcome measure that was controlled for total brain volume and brain growth during the observation period, we used the ratio of ipsilateral over contralateral volume of intact tissue (Ri):

$$R_i = \frac{V_{i,i}}{V_{i,c}}$$

**Eq 1. Ratio of intact ipsilateral volume.** $V_{i,i}$: Volume of intact (normointense) ipsilateral volume, $V_{i,c}$: Volume of intact (normointense) contralateral volume.

DTI analyses were performed with tools of the FMRIB software library (FSL version 4.1.4, Oxford Centre for Functional MRI of the Brain, UK; http://www.fmrib.ox.ac.uk/fsl) [12,13]. Images were pre-processed to reduce image artifacts due to motion and eddy current distortions by affine transformation and co-registration of the diffusion encoded images to the b0-images. Images with severe ghosting artifacts were excluded from further analyses (2 HI +PBS on day 7 after HI and 1 HI+DMSO on day 20 after HI). FDT version 2.0 (part of FSL) was used to fit a voxelwise diffusion tensor model to the DTI data [14]. Maps for the fractional anisotropy (FA), mean diffusivity, radial diffusivity and axial diffusivity were created. A ROI was manually drawn in the body of the corpus callosum and mean FA, mean, radial and axial diffusivity were calculated.

## Functional testing

All testing was performed between 10:00 a.m. and 16:00 p.m. in a subgroup of animals (sham n = 6, HI+MEL n = 7, HI+DMSO n = 7, HI+PBS n = 5). The tests were carried out by the same individual who was blinded to the experimental grouping at all times.

The cylinder rearing test [15] was performed at 15 and 36 days after HI. Animals were placed in an age-appropriate glass cylinder, which was surrounded by 3 mirrors, and video recorded for 5 minutes. The video was reviewed frame by frame and number of contacts between the forepaw and wall were counted and registered as left, right or both (if both paws made contact simultaneously). Only contacts involving all digital pads and the palmar pads were registered. Forepaw preference ratio was calculated for each limb and for both limbs by dividing number of contacts with the respective limb by total number of contacts.

The novel object recognition test [16] was performed at 12 and 41 days after HI. During the first 2 days, the animal was placed in an empty heated PVC box (48 x 96 x 46 cm) and allowed to explore it freely for 10 minutes (habituation). On day 3, the animal was placed in the box together with 2 identical blue conical plastic objects for 5 (12 days after HI) or 3 (41 days after HI) minutes (familiarization). After 1 hour (short term test) one object was replaced by an ovoid plastic object and the animal was allowed to explore the box for 3 minutes. This was

repeated after 24 hours (long-term test) with a cylindrical white object with a red base. Animals were video recorded during familiarization, short-term testing and long-term testing. Exploration was defined as either orientation of the animal's snout towards the object, sniffing of the object, touching the object with the snout or rearing the object, and timed for novel and familiar object. Exploration time (ET) was calculated as time exploring novel object plus time exploring familiar object. Difference in exploration time (D1) was calculated as time exploring new object minus time exploring old object [17]. Animals with ET < 2 s were excluded for analysis of D1.

## Histology and immunohistochemistry

The animals were euthanized 43 days after HI and the brains were dissected and embedded in paraffin. Coronal sections corresponding to -3.25 mm from Bregma [18] were cut and stained with hematoxylin-eosin (H&E, Haematoxylin, CellPath/Chemi-Teknik, Oslo, Norway; Erythrosine B, Sigma-Aldrich, Oslo, Norway); with luxol fast blue (CellPath/Chemi-Teknik, Oslo, Norway) to detect demyelination; and with anti-glial fibrillary acid protein (anti-GFAP Z0334, Dako, Oslo, Norway) as a marker for reactive astrocytosis in 23 animals (12 sham and 11 HI). For the GFAP immunostaining, the sections were incubated in labelled polymer horseradish peroxidase anti-Rabbit and diaminobenzodine (Dako, Oslo, Norway). All slices were examined using a light microscope (Olympus BX41, Tokyo, Japan).

H&E sections were semi-quantitatively scored at x20 for neuronal cell loss (3 = > 67% stained, 2 = 33–67% of cells stained, 1 = < 33% of cells stained and 0 = no stained cells) in 8 ipsilateral areas, including the somatomotor (medial) cortex, somatosensory cortex, auditory cortex, CA1-3 regions of the hippocampus, thalamus and caudate putamen. The total histology score (0–12) was the sum of the average score within the cortical areas, the average score of the CA1-3 regions and the scores in the thalamus and putamen. Two histology scores are missing because of problems with the histological preparations (2 HI+MEL). The mean histology score was calculated for all HI animals, and each HI animal placed in the group "severe" if its histology score was below the mean and "mild" if the score was above.

The thickness of the body of the corpus callosum was measured on the luxol fast blue stained slices at x10 using labSens (version 1.3; Olympus Soft Imaging Solutions, Hamburg, Germany).

## Statistical analysis

Analyses were performed in R 3.3.3 R Core Team, 2017.Variables were tested for normality using qq-plots, histograms and a Shapiro-Wilk test. When comparing two groups, Student's t-test was used for normally distributed variables and a Mann-Whitney U test was used for variables not normally distributed. For paired samples, a paired t-test or paired MWU was used. When comparing more than two groups, one-way ANOVA or Kruskal-Wallis rank sum test was used. Differences were considered significant when the two-sided p value was $\leq 0.05$. Associations between continuous variables were assessed using Pearson's product moment correlation coefficient for normally distributed variables with an assumed linear correlation based on scatterplots, or Spearman's rho if these conditions were not met. The frequency distribution of experimental subjects into mutually exclusive groups was compared using a Fisher exact test. Graphical plots were produced using the ggplot2 package for R [19].

We found no differences between the treatment groups in the sham animals and therefore we considered sham as one group in all analyses. Furthermore, the HI+DMSO were not different from the HI+PBS animals and were clustered to HI+Vehicle to increase statistical power.

## Results

### Treatment effects day 1 after HI

HI animals had lower mean ADC values in several ipsilateral gray matter areas compared to sham animals (Fig 2A). Among both melatonin and vehicle treated animals there were clusters of animals with higher ADC than sham in the external capsule. However, the mean ADC was not different between melatonin and vehicle treated animals in any area.

The volumes of tissue with increased signal intensity on $T_2$-weighted images in the ipsilateral hemisphere (indicating injured tissue with edema) was higher in HI animals compared to sham. On average, melatonin treated animals tended to have lower volumes of injured tissue

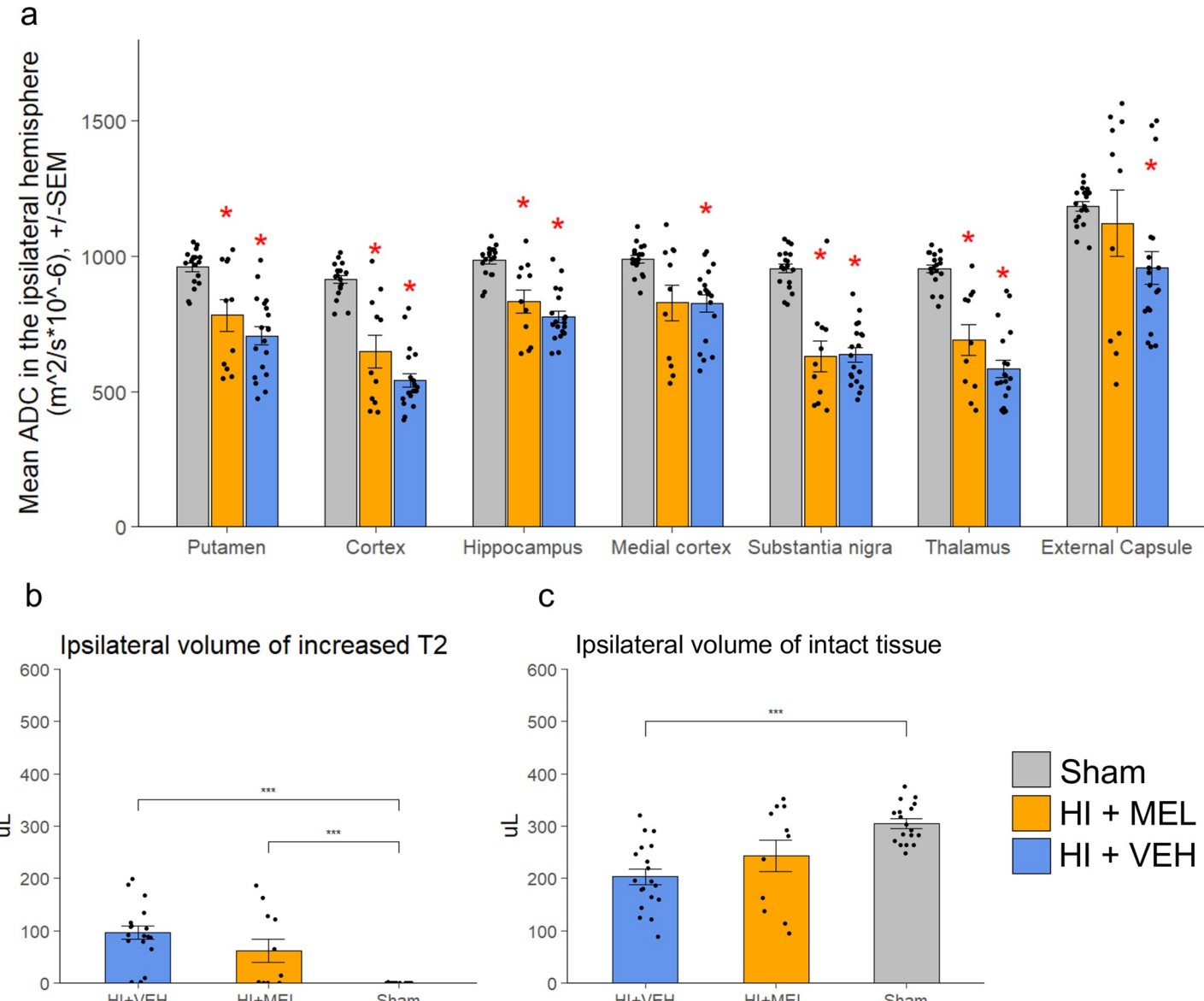

**Fig 2. MRI metrics 1 day after HI.** ADC and $T_2$-weighted images were acquired one day after HI. a: ADC values were measured in different areas in the ipsilateral hemisphere and compared between treatment groups. *: Significantly different from sham ($p < 0.05$). b and c: $T_2$-weighted images were used to measure b) ipsilateral volume of increased signal and c) ipsilateral volume of normointense tissue (***: $p < 0.001$. *: $p < 0.05$). Error bars represent SEM.

than vehicle treated, but this was not statistically significant (p = 0.156) (Fig 2B). Furthermore, vehicle treated animals had lower volumes of normally looking tissue than sham animals, whereas such volumes among the melatonin treated animals were in between that of vehicle and sham animals and not significantly different from either (Fig 2C).

### Long-term lesion volume development

On day 1 the mean Ri (volume of intact brain tissue in ipsilateral over contralateral hemisphere, Eq 1) was lower in vehicle treated animals compared to sham animals, whereas the mean Ri for melatonin treated animals was not significantly different from either sham or vehicle treated animals (Fig 3A).

Ri was reduced over time among the HI animals, with the most dramatic decrease happening from day 1 to day 7 (Fig 3A–3D). From day 7, the mean Ri of melatonin treated animals was lower than sham. At no time point was there any differences in mean Ri between melatonin and vehicle treated HI animals.

There was a large variation in Ri among the HI animals, showing a clustering of HI animals into two clusters on day 1 with either 1) high Ri similar to sham or 2) low Ri. Animals were similarly separated on day 43 with cluster 1 having high to medium Ri and cluster 2 having low Ri (Fig 3E).

Animals in cluster 1 were also characterized by higher ADC in the ipsilateral external capsule compared to sham, whereas animals in cluster 2 had lower ADC than sham on day 1 after HI. The two clusters also differed in almost all other outcome measurements at all timepoints with cluster 1 showing less severe signs of injury (S1 Table).

There was a skewed distribution of melatonin and vehicle treated animals in the two clusters (Table 1) with a higher fraction of melatonin treated animals in cluster 1 (55%) compared to vehicle treated animals (15.8%), (p = 0.042, Fisher's exact test).

### White matter injury development in the corpus callosum

On average there were no differences in FA, mean or radial diffusivity between the melatonin and Vehicle treated HI animals at any time-point (Fig 4). However, while the axial diffusivity was lower in the vehicle treated than in sham animals at all time-points (p < 0.01), it was only lower on day 7 after HI in the melatonin treated animals compared to sham (p<0.005).

Luxol fast blue staining that showed increasing demyelination and thinning of the corpus callosum with increasing injury severity (Fig 4G–4I). The measured thickness of corpus callosum was correlated to intact ratio ($R^2$ = 0.8, p<0.001), FA ($R^2$ = 0.62, p<0.001, Fig 4F) and radial diffusivity ($R^2$ = 0.47, p<0.001) in the body of corpus callosum on day 43 after HI. Although there was a slight tendency towards thicker corpus callosum among melatonin treated than vehicle treated HI animals (Fig 4E), no statistically significant difference was found between the groups (p = 0.39).

### Functional outcome

We did not find any effect of treatment on any of the functional test parameters used in this study (Fig 5). HI animals had a higher ratio of rearings using the right forepaw compared to sham at 15 (p 0.023) and 36 (p 0.003) days after HI. The ratio remained unchanged over time in both groups. In the novel object recognition test, preference for the new object was found only at the late time-point. Total exploration time (ET) was lower in HI animals compared to sham at the 1 hour test at 14 days after HI (p 0.010) and at the 24 hours test at 43 days after HI (p 0.004).

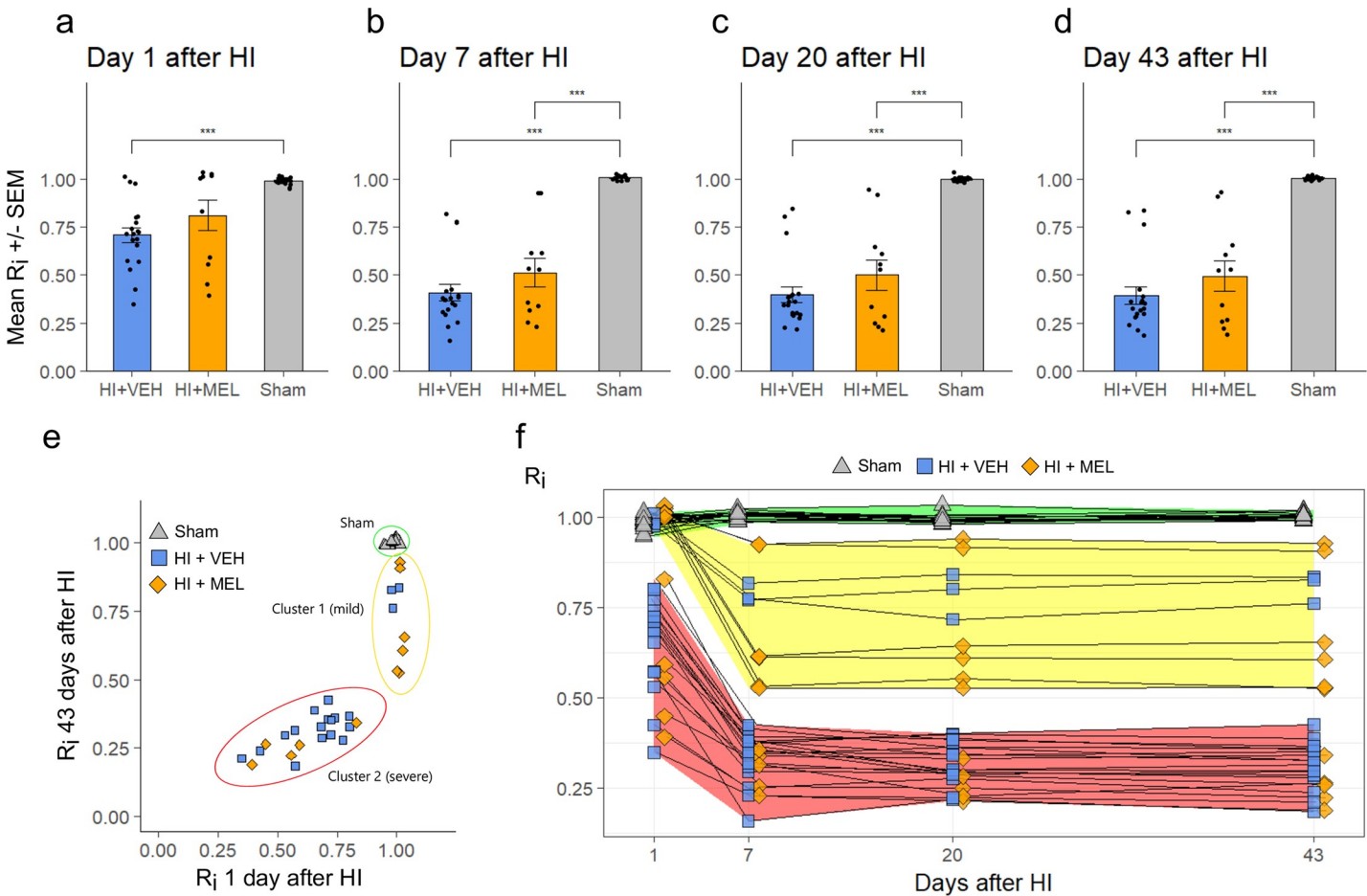

**Fig 3. Injury development on MRIs.** Intact ratio ($R_i$) was defined as the ratio of intact tissue volume in the ipsilateral hemisphere to the intact tissue volume in the contralateral hemisphere, and was measured at 1 (a), 7 (b), 20 (c) and 43 (d) days after HI. a-d: Mean $R_i$ for each treatment group at different time points, +/- SEM. ***: $p < 0.001$ (MWU). e: $R_i$ for individual animals over time.

## Histology 43 days after HI

Histology scores were not different between treatment groups. The total histology score correlated well with the $T_2$ volumetric measurements in the ipsilateral hemisphere at all time-points (Fig 6C).

When looking at H&E and GFAP stained sections, the HI animals in cluster 1 showed mildly injured brains with no tissue loss and only gliosis in the hippocampus and to some

**Table 1. Distribution of animals in injury severity clusters.**

|  |  | Sham |  | Mild |  | Severe |  |
| --- | --- | --- | --- | --- | --- | --- | --- |
|  |  | n | % | n | % | n | % |
| Sham | n = 18 | 18 | 100.0% | - |  | - |  |
| HI + MEL | n = 11 | - |  | 6 | 54.5% | 5 | 45.5% |
| HI + VEHICLE | n = 19 | - |  | 3 | 15.8% | 16 | 84.2% |

Distribution of melatonin treated and vehicle treated animals in the clusters base on Ri on day 1 and 43 after HI. Cluster 1 showed a more mild injury development than cluster 2.

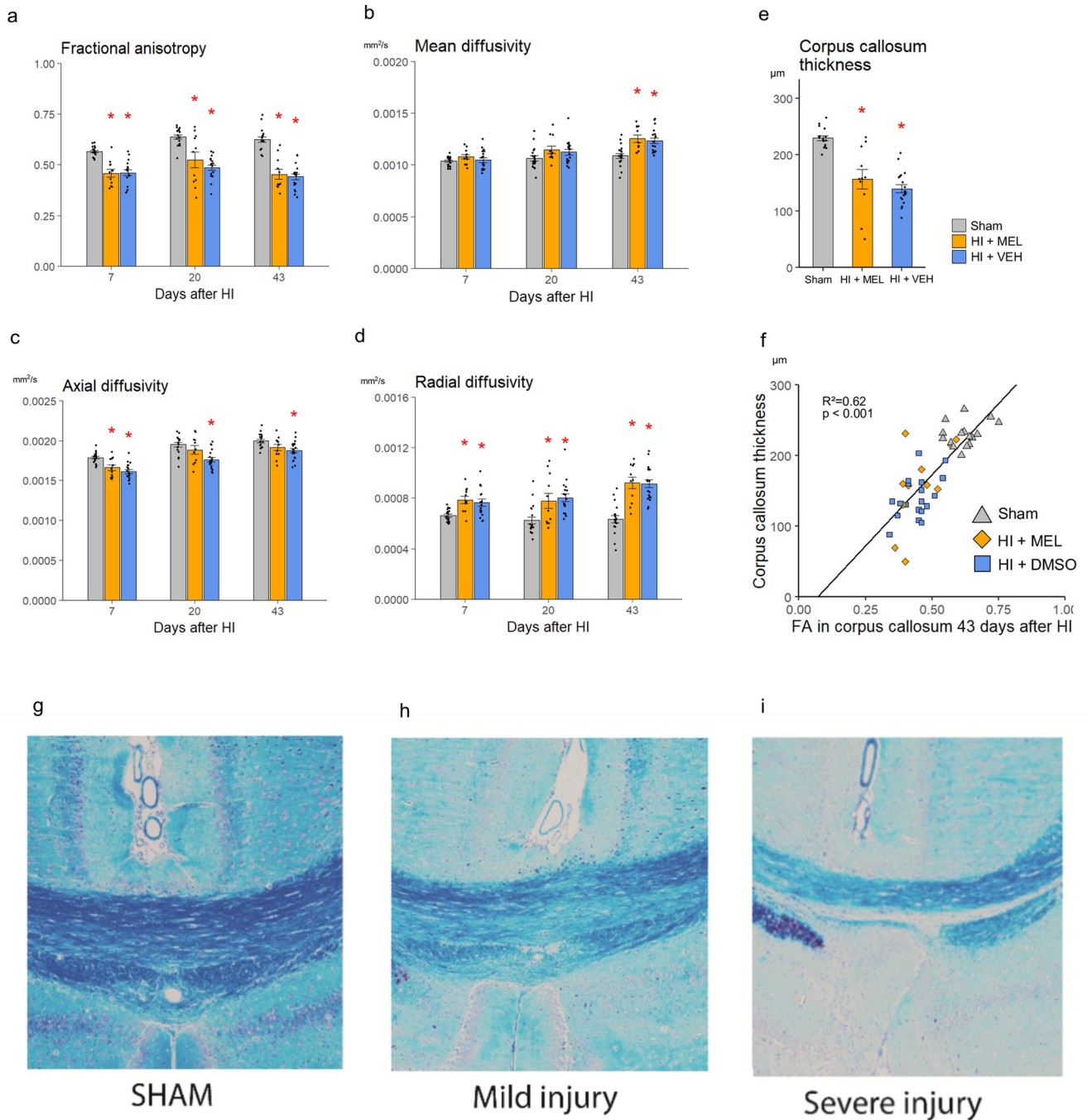

**Fig 4. Diffusion tensor imaging and histology in corpus callosum.** Diffusion tensor imaging (DTI) was performed at days 7, 20 and 43 after HI. a-d: DTI measurements of the corpus callosum over time. g-i: Luxol fast blue stained histological slides of the corpus callosum 43 days after HI, demonstrating increasing thinning with increased injury severity. Error bars indicate SEM. *: Significantly different from sham, $p < 0.05$.

extent in the lateral cortex (Fig 6). In comparison, the HI animals in cluster 2 were severely injured with intact medial cortex but major tissue loss in the lateral cortex and gliosis and/or tissue loss in the hippocampus, putamen and thalamus.

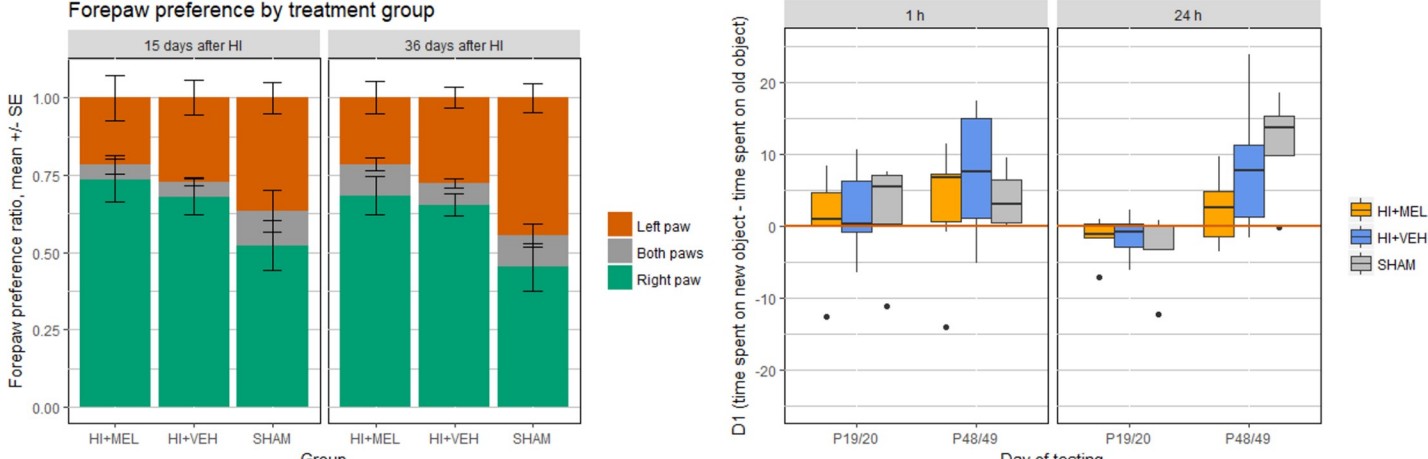

**Fig 5. Functional tests.** Performance on the cylinder rearing test and the object recognition test at 2 and 5–6 weeks after HI. (a) Results from the cylinder rearing test. The HI animals had a right forepaw preference compared to sham at 2 and 5 weeks after HI. Results are presented in mean ± SE. (b) Boxplots showing the results from the novel object recognition test. The difference in exploration time for the novel and old object, D1, was used as a measure of novel object preference. A positive D1 value indicates a preference for the novel object and a negative D1 value indicates a preference for the old object. The animals showed a preference for the novel object only at 6 weeks after HI. HI, hypoxia-ischemia; MEL, melatonin; sham, sham-operated animals; VEH, vehicle-treated.

## Sex differences

There was an unequal distribution of sexes among the HI animals with more females in the HI +MEL group (9 females versus 2 males) and more males in the HI+vehicle (6 females versus 13 males). (Table 2). When subdividing into severity, the distribution of sexes reflected the overall distribution in the treatment group. Further, there were no differences found between the sexes in any outcome measure at any time-point (S2 Table).

## Discussion

### Transient effect of melatonin treatment on long-term brain injury development

In this study, we found only subtle differences between animals treated with melatonin and those receiving vehicle treatment that were in favor of melatonin treatment. Only on day 1 was there any tendency towards treatment effects when looking at the average measured intact tissue volumes and ADC values in the external capsule on MRI. This could indicate a modest treatment effect day 1 after HI. However, on average no differences were seen at later time-points for any outcome measures (brain injury volumes, DTI parameters, corpus callosum thickness, histology, functional outcome) when looking at the animals at the group level. This indicates that any effects of the melatonin treatment on day 1 could be transient.

The large variability in extent of injury in all HI treatment groups made it challenging to compare treatment effects on a group level in this study. However, when we looked at the distribution of the observed lesion sizes day 1 after HI and at the end of the study, the HI animals appeared to be clustered into two groups: One with apparently mild or non-existing injuries 1 day after HI (cluster 1) and one with apparently severe injuries 1 day after HI (cluster 2). These clusters were different throughout the whole observational period in all observational domains (volumetry, diffusion MRI, histology and functional tests), with cluster 2 showing a more severe phenotype.

It is well known that the variation in injury extent in the Vannucci model is high [20]. This was exemplified by the HI+vehicle group, in which 15.8% was in cluster 1 which developed a

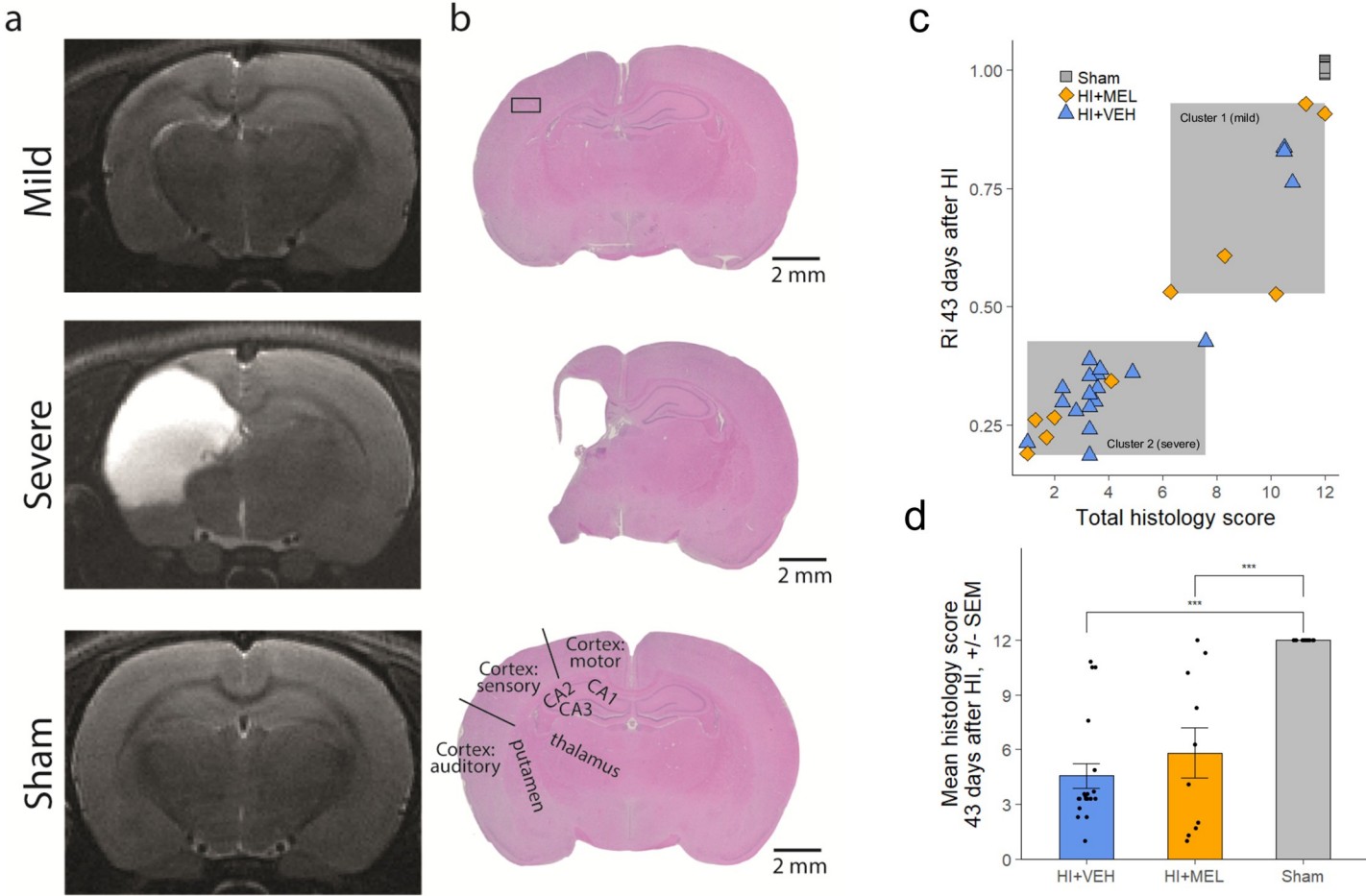

**Fig 6. Brain injury at 43 days after HI on MRI and histology.** Histology. (a) Representative T2-w images acquired on day 43 after HI and (b) corresponding histological slices stained with H&E in the different injury groups. (c): Scatterplot of total histological score vs ipsilateral intact volume as measured on MRI, demonstrating division into two clusters and good correlation (Spearman's rho = 0.944). (d) The total histology score did not differ between HI+MEL and HI+vehicle animals. H&E; hematoxylin-eosin; HI, hypoxia-ischemia; MEL, melatonin; sham, sham-operated animals.

mild injury and 84.2% was in cluster 2 with a severe injury. Although we do not have information about the extent of the initial injury (before treatment), one should expect the same distribution in the HI+MEL group if no treatment effect is assumed. However, only 45.4% of the melatonin treated animals had severe injuries. This indicates an absolute risk reduction for severe injury of 38.7%, with a number needed to treat of 2.6 for a better radiological and histological outcome. This is in line with previous reports [21–23].

There were also interesting differences in the injury trajectories among the animals in cluster 1: About half of the melatonin treated animals in cluster 1 developed a worse injury than

**Table 2. Distribution of animals according to sex, injury severity and treatment group.**

|  |  | Sham n = 18 | | Mild n = 9 | | Severe n = 21 | |
|---|---|---|---|---|---|---|---|
|  |  | **Females** | **Males** | **Females** | **Males** | **Females** | **Males** |
| Sham | n = 18 | 8 | 10 | - | - | - | - |
| HI + MEL | n = 11 | - | - | 5 | 1 | 4 | 1 |
| HI + vehicle | n = 19 | - | - | 1 | 2 | 5 | 11 |

the other half of melatonin treated animals and the vehicle treated animals in cluster 1. One could speculate that these melatonin treated animals had a transient effect of treatment on day 1. Reasons why these animals were not fully protected might be insufficient effect of a dosage of 10 mg/kg melatonin to protect the brain from secondary injury. Furthermore, three injections within the first 25 hours could be too few to exert long-term neuroprotective effects, even though a similar treatment regimen has been reported to provide favorable short term outcomes [22,24]. Most studies that examined the effect of melatonin after neonatal HI brain injury have only evaluated short-term effects (until 1 week after HI) or used indirect prognostic biomarkers [25–28]. However, this study emphasizes the importance of a long-term follow-up to investigate the effects of neuroprotective agents after HI.

It has previously been shown that melatonin may have protective effects on white matter [28]. Although there was a slight tendency towards thicker corpus callosum among melatonin treated animals in our study, we could not establish any specific protection of white matter maturation related to melatonin treatment as the thickness and DTI parameters of the corpus callosum was foremost correlated to the overall brain injury and not to treatment.

A disadvantage with the design of this study is that MRI was not performed before melatonin treatment, making it difficult to isolate treatment effects. ADC maps can be obtained shortly after HI and give some information about the severity of the insult, but the tissue changes needed to completely predict the severity are not manifested until 24–48 h after injury [29]. Further, performing MRI before treatment on all animals in a whole litter is not practically feasible in this type of study due to the time it takes for each MRI scan. Several serum biomarkers have been proposed but efforts should be made to validate these biomarkers against well-established outcome measures [30]. As such, there are currently no reliable biomarkers that can distinguish severity of HI early enough to be analyzed before treatment needs to be administered.

## Sex differences in melatoninergic neuroprotection

A limitation of the present study is the unequal distribution of the sexes in the treatment groups as a result of randomization, with more females in the melatonin-treated group and more males in the vehicle-treated group. Although we did not find any significant differences between male and female rats in any outcome measure and the distribution of sexes was equal in both injury severity clusters for each treatment condition, we cannot exclude any sex-dependent effects. It is known that male rats have worse brain injuries, reduced myelination and more behavioral deficits after perinatal HI compared to females and this is linked to sex-specific hormones, metabolism and activated pathological pathways [3,31]. Furthermore, Tai et al. found neuroprotective effects after stroke at lower melatonin dosing regimens in female (adult) Sprague-Dawley rats [32]. This might be caused by synergistic anti-oxidant and anti-inflammatory effects of estrogen and melatonin as well as possible effects of estrogen on the function and density of melatonin receptors. However, there are no sex differences in plasma estrogen levels at P7 yet [33]. In conclusion, this study does not provide evidence against recommending melatonin to males and females, although further studies on melatoninergic neuroprotection after neonatal hypoxic-ischemic brain injury should take sex differences into account, and use randomization methods that prevent skewed grouping.

## Conclusion

Melatonin treatment during the first day after HI in neonatal rats resulted in more mild versus severe brain injury as measured on MRI day 1 after HI. However, about half the melatonin treated animals that showed signs of mild injury on day 1 developed a more moderate injury

over time indicating a transient effect of melatonin treatment. These findings suggest that 3 post-HI injections with 10 mg/kg melatonin during the first day after HI may have some protective effects on the most mildly injured animals, but are not sufficient to fully protect the brain from delayed injury after HI. Further research should focus on the optimization of melatonin treatment regimens, early reliable biomarkers for HI severity as well as possible sex-dependent neuroprotective effects of melatonin.

## Supporting information

**S1 Table. Results for clusters of animals with mild and severe injuries.**
(DOCX)

**S2 Table. Results for male and female HI rats.**
(DOCX)

## Acknowledgments

The MRI was performed at the MR Core Facility at NTNU. The authors would like to thank the personnel at the Comparative Medicine Core Facility for their assistance and professor Sverre Helge Torp and staff engineer Ingunn Nervik at the Cellular & Molecular Imaging Core Facility for the histological sections and analysis.

## Author Contributions

**Conceptualization:** Hester Rijkje Berger, Tora Sund Morken.

**Data curation:** Axel K. G. Nyman.

**Formal analysis:** Hester Rijkje Berger, Axel K. G. Nyman, Marius Widerøe.

**Funding acquisition:** Marius Widerøe.

**Investigation:** Hester Rijkje Berger, Axel K. G. Nyman.

**Methodology:** Hester Rijkje Berger, Tora Sund Morken.

**Project administration:** Marius Widerøe.

**Resources:** Marius Widerøe.

**Software:** Marius Widerøe.

**Supervision:** Tora Sund Morken, Marius Widerøe.

**Visualization:** Axel K. G. Nyman.

**Writing – original draft:** Hester Rijkje Berger, Axel K. G. Nyman, Tora Sund Morken, Marius Widerøe.

**Writing – review & editing:** Hester Rijkje Berger, Axel K. G. Nyman, Tora Sund Morken, Marius Widerøe.

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
