## [Decision Letter · Decision Letter 0]

11 Aug 2019

PONE-D-19-16889

Transient effect of melatonin treatment after neonatal hypoxic-ischemic brain injury in rats

PLOS ONE

Dear Dr. Widerøe,

Thank you for submitting your manuscript to PLOS ONE. After careful consideration, we feel that it has merit but does not fully meet PLOS ONE’s publication criteria as it currently stands. Therefore, we invite you to submit a revised version of the manuscript that addresses the points raised during the review process.

We would appreciate receiving your revised manuscript by Sep 25 2019 11:59PM. To enhance the reproducibility of your results, we recommend that if applicable you deposit your laboratory protocols in protocols.io, where a protocol can be assigned its own identifier (DOI) such that it can be cited independently in the future. For instructions see: http://journals.plos.org/plosone/s/submission-guidelines#loc-laboratory-protocols

We look forward to receiving your revised manuscript.

Kind regards,

Olivier Baud, MD, PhD

Academic Editor

PLOS ONE

Journal Requirements:

1. We note that you have stated that you will provide repository information for your data at acceptance. Should your manuscript be accepted for publication, we will hold it until you provide the relevant accession numbers or DOIs necessary to access your data. If you wish to make changes to your Data Availability statement, please describe these changes in your cover letter and we will update your Data Availability statement to reflect the information you provide.

Reviewers' comments:

Reviewer's Responses to Questions

**Comments to the Author**

1. Is the manuscript technically sound, and do the data support the conclusions?

Reviewer #1: Partly

2. Has the statistical analysis been performed appropriately and rigorously? 

Reviewer #1: Yes

3. Have the authors made all data underlying the findings in their manuscript fully available?

Reviewer #1: Yes

4. Is the manuscript presented in an intelligible fashion and written in standard English?

Reviewer #1: Yes

5. Review Comments to the Author

Reviewer #1: Berger et al. describe the effects of melatonin (MEL) treatment in a model of hypoxia-ischemia (HI) in the P7 rat.

While this is an interesting research topic (considering the safety of the molecule and clinical settings) we feel that the manuscript need several improvements before it can be accepted.

- In the introduction consider more recent works on melatonin in clinical settings: Biran et al. 2019 Int J mol Sci; Colella et al., 2016, Early Hum Dev 102: 1-3; Biran et al., 2014, 56: 717-723.

- Again, in the discussion, consider several papers reporting that MEL promotes myelination with oligodendroglial maturation (Olivier et al., PlosOne 2009, 4, e7128) and decreases white matter inflammation (Villapol et al., Ped Res 2011, 69:51-55) in neonatal rats after cerebral injury.

- As in your study you also shown that HI induced demyelination, it could has been interesting to measure the thickness of myelin (in the corpus callosum) in the 3 groups of animals, and to evaluate whether MEL was associated to an increase in the density of mature oligodendrocytes.

- WM injury is the underlying cause of motor and cognitive disability in injuried babies that suffer HI and/or ischemia. Evaluation of sensorimotor performance and motor coordination by using the pole test could have been a good test to see (or not) an effect of MEL on HI (namely in mild injured animals).

- Concerning no sex-difference in this study allows to concider MEL as a broad spectrum treatment.

6. PLOS authors have the option to publish the peer review history of their article (what does this mean?). If published, this will include your full peer review and any attached files.

Reviewer #1: Yes: Christiane CHARRIAUT-MARLANGUE

---

## [Author Response · Author response to Decision Letter 0]

24 Oct 2019

Response to reviewers

We would like to thank the reviewers for their time spent reviewing the manuscript and for their valuable comments. We have responded to all issues and questions raised the by the editor and reviewer and made necessary changes to the manuscript and figures to accommodate your critique. See specific answers to each point raised below. 

Journal Requirements:

 We have gone over the manuscript and made sure it complies with the PLOS style requirements.

1. We note that you have stated that you will provide repository information for your data at acceptance. Should your manuscript be accepted for publication, we will hold it until you provide the relevant accession numbers or DOIs necessary to access your data. If you wish to make changes to your Data Availability statement, please describe these changes in your cover letter and we will update your Data Availability statement to reflect the information you provide.

The data has been deposited in the repository and the DOI is provided in the cover letter.

We have now included the data in a supplementary table (S2 Table) and included a reference to that in the manuscript.

Reviewer #1: 

Berger et al. describe the effects of melatonin (MEL) treatment in a model of hypoxia-ischemia (HI) in the P7 rat.

While this is an interesting research topic (considering the safety of the molecule and clinical settings) we feel that the manuscript need several improvements before it can be accepted.

- In the introduction consider more recent works on melatonin in clinical settings: Biran et al. 2019 Int J mol Sci; Colella et al., 2016, Early Hum Dev 102: 1-3; Biran et al., 2014, 56: 717-723.

Thank you for this comment. References to the suggested work has been included in the introduction section in the manuscript.

- Again, in the discussion, consider several papers reporting that MEL promotes myelination with oligodendroglial maturation (Olivier et al., PlosOne 2009, 4, e7128) and decreases white matter inflammation (Villapol et al., Ped Res 2011, 69:51-55) in neonatal rats after cerebral injury.

Thank you for bringing our attention to these papers. A reference to Villapol et al has been added to the discussion regarding short term effects. The study by Olivier et al provides compelling evidence of mechanisms of melatoninergic neuroprotection. We feel, however, that the choice of animal model (e17 uterine artery ligation) makes the result more relevant to a slightly different clinical problem (diffuse white matter injury in very preterm infants) than we attempt to address (perinatal hypoxia-ischemia).

- As in your study you also shown that HI induced demyelination, it could has been interesting to measure the thickness of myelin (in the corpus callosum) in the 3 groups of animals, and to evaluate whether MEL was associated to an increase in the density of mature oligodendrocytes.

Thank you for this suggestion. We have now included the measurements of corpus callosum in all three groups in the results (Figure 4, and results section under white mater injury development in cc) and commented on this in the discussion. The results indicate a slight tendency towards better preservation of the cc among MEL animals, but the difference was not near statistical significance and cc thickness was foremost correlated with the overall brain injury. Unfortunately, we are not able to perform further immunohistochemistry to investigate further the oligodendrocyte density and maturity.

- WM injury is the underlying cause of motor and cognitive disability in injuried babies that suffer HI and/or ischemia. Evaluation of sensorimotor performance and motor coordination by using the pole test could have been a good test to see (or not) an effect of MEL on HI (namely in mild injured animals).

Thank you for this suggestion. We chose to evaluate sensorimotor performance using the cylinder rearing test based on extensive use of this test in previous neonatal HI research and ease of comparison to previous work. The test is sensitive to unilateral lesions in all parts of the corticospinal tract (Schallert T, Fleming SM, Leasure JL, Tillerson JL, Bland ST Neuropharmacology. 2000;39:777–87.). Normalization of test performance follows restoration of corticospinal tract function in the neonatal mouse (van Velthoven CTJ, van de Looij Y, Kavelaars A, Zijlstra J, van Bel F, Huppi PS, et al Ann Neurol. 2012;71:785–96.). A disadvantage of this test is that it depends on lateralization of sensorimotor function, but since the Vannucci model is (mostly) unilateral, the test is well suited to this model in our opinion. The addition of a less specific test such as the pole test could perhaps increase the chance of discovering non-lateralized sensorimotor dysfunction. We will keep this in mind when designing experiments in the future.

- Concerning no sex-difference in this study allows to concider MEL as a broad spectrum treatment.

Thank you for this comment. We agree that this study does not provide evidence against recommending melatonin to either males or females. However, as discussed under “Sex differences in melatoninergic neuroprotection”, we do not feel that our study can answer the question of sex-dependent effects with a high degree of confidence, due to the skewed grouping of males and females in the treatment groups. A sentence addressing this has been added to the discussion.

---

## [Editor Report · Decision Letter 1]

13 Nov 2019

Transient effect of melatonin treatment after neonatal hypoxic-ischemic brain injury in rats

PONE-D-19-16889R1

Dear Dr. Widerøe,

We are pleased to inform you that your manuscript has been judged scientifically suitable for publication and will be formally accepted for publication once it complies with all outstanding technical requirements.

With kind regards,

Olivier Baud, MD, PhD

Academic Editor

PLOS ONE
---

## [Editor Report · Acceptance letter]

3 Dec 2019

PONE-D-19-16889R1 

Transient effect of melatonin treatment after neonatal hypoxic-ischemic brain injury in rats 

Dear Dr. Widerøe:

I am pleased to inform you that your manuscript has been deemed suitable for publication in PLOS ONE. Congratulations! Your manuscript is now with our production department. 

With kind regards,

on behalf of

Pr. Olivier Baud 

Academic Editor

PLOS ONE